# BalancEdit: Dynamically Balancing the Generality-Locality Trade-off in Multi-modal Model Editing

Dongliang Guo [1]  Mengxuan Hu [1]  Zihan Guan [1]  Thomas Hartvigsen [1]  Sheng Li [1]

## Abstract

Large multi-modal models inevitably decay over time as facts update and previously learned information becomes outdated. Traditional approaches such as fine-tuning are often impractical for updating these models due to their size and complexity. Instead, direct knowledge editing within the models presents a more viable solution. Current model editing techniques, however, typically overlook the unique influence ranges of different facts, leading to compromised model performance in terms of both generality and locality. To address this issue, we introduce the concept of the generality-locality trade-off in multi-modal model editing. We develop a new model editing dataset named OKEDIT, specifically designed to effectively evaluate this trade-off. Building on this foundation, we propose **BalancEdit**, a novel method for balanced model editing that dynamically achieves an optimal balance between generality and locality. BalancEdit utilizes a unique mechanism that generates both positive and negative samples for each fact to accurately determine its influence scope and incorporates these insights into the model's latent space using a discrete, localized codebook of edits, without modifying the underlying model weights. To our knowledge, this is the first approach explicitly addressing the generality-locality trade-off in multi-modal model editing. Our comprehensive results confirm the effectiveness of BalancEdit, demonstrating minimal trade-offs while maintaining robust editing capabilities. Our code and dataset are available at https://github.com/donglgcn/BalancEdit/tree/MMOKVQA.

[1]University of Virginia, Charlottesville, USA. Correspondence to: Sheng Li <shengli@virginia.edu>.

*Proceedings of the 42nd International Conference on Machine Learning*, Vancouver, Canada. PMLR 267, 2025. Copyright 2025 by the author(s).

## 1. Introduction

Large multi-modal models (Zhu et al., 2023; Radford et al., 2021; Li et al., 2023; Liu et al., 2023a; Rombach et al., 2022) have recently brought about significant advancements in artificial intelligence, demonstrating impressive results in tasks such as Visual Question Answering (VQA) (Antol et al., 2015). However, these models are susceptible to issues like hallucination (Rawte et al., 2023) and fact alteration (De Cao et al., 2021; Guo et al., 2024; Zhu et al., 2024). After deployment, these models may generate numerous errors, leading to potential problems like the propagation of hate speech (Guan et al., 2025b;a; Hu et al., 2025; 2024) or the dissemination of outdated factual information. Given these challenges, it is critical to continually update and maintain these large multi-modal models to ensure their accuracy and relevance.

While retraining or fine-tuning can update a model's knowledge, it is often infeasible to frequently edit individual facts due to the high computational costs involved. Fortunately, model editing techniques (Hartvigsen et al., 2024; Mitchell et al., 2021; Zheng et al., 2023) provide a promising approach to implementing cost-effective, targeted updates to large pretrained models. These techniques typically involve injecting new layers or modifying weights to alter the knowledge embedded in language models. A successful edit generally exhibits three characteristics (Mitchell et al., 2021; Huang et al., 2023): **reliability**, which ensures the output changes to the target answer for the same question; **locality**, which leaves unrelated knowledge and outputs unchanged; and **generality**, which produces the correct answer for all questions within the influence scope. As illustrated in Fig. 1, each fact has its own influence scope. For instance, if we wish to edit the name of a specific cat, the influence scope would be confined to that particular cat. If we aim to edit the name of a cat breed, the influence scope would extend to all cats within that breed. However, if we intend to edit the name of a species, the influence scope would encompass all cats. Consequently, we should consider each fact individually and dynamically to determine the appropriate influence scope.

However, current model editing techniques often overlook the dynamic nature of the influence scope. Some meth-

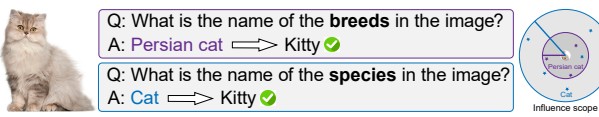

*Figure 1.* Illustration of various influence scope

| | Original Image | Related Image | Unrelated Image | Black Image |
|---|---|---|---|---|
| Question | | What brand is this computer? | | |
| Target | | hp → lenovo | | |
| Base | hp | hp | dell | black |
| IKE | lenovo | lenovo | lenovo | lenovo |
| MEND | lenovo | lenovo | lenovo | lenovo |
| GRACE | lenovo | hp | dell | black |
| Ours | lenovo | lenovo | dell | black |

*Table 1.* An example of generality-locality trade-off. Red color means the false prediction and Green color indicates the correct prediction

ods treat all influence scopes as if they are large and uniform, while others focus solely on a specific edit. For instance, IKE (Zheng et al., 2023) employs in-context learning to edit knowledge, using the closest piece of knowledge as a prompt to guide the language model. This approach causes the language model to rephrase the nearest fact, resulting in an oversized influence scope. Conversely, GRACE (Hartvigsen et al., 2024), a lifelong model editing method, assumes that each edit has a small and similar influence range, leading to limited generality. Consider an example where we aim to edit a "fact" that HP computers have been renamed Lenovo, as shown in Table 1. Ideally, model editing should update the answer from HP to Lenovo whenever it encounters an image of a HP computer, while leaving the answer unchanged for other brands. However, existing model editing techniques, such as IKE (Zheng et al., 2023) and MEND (Mitchell et al., 2021), may achieve the target edit but neglect the influence scope, inadvertently editing other brands as well. Even when presented with a black image, these models may still output the new answer, leading to hallucination. On the other hand, while GRACE (Hartvigsen et al., 2024) maintains the backbone model's answer for unrelated images, it fails to edit the knowledge to the desired scope. These observations suggest that existing multi-modal model editing methods struggle to dynamically adjust the influence scope of a knowledge edit, and to balance generality and locality effectively.

To address this issue, we first create a dataset designed to evaluate the trade-off between generality and locality in model editing techniques. We then introduce an efficient multi-modal model editing method named **BalancEdit**, which dynamically balances this trade-off with minimal computational costs. Specifically, we incorporate an adapter into a chosen layer of a vision language model without altering its weights. This adaptor modifies layer-to-layer transformations for select inputs. By caching embeddings for input edits and the updated knowledge transformation layer, BalancEdit functions as a codebook where edits are stored. To strike a balance between generality and locality, we generate the corresponding positive and negative samples for each edit. The model's semantic similarity in its latent space can be visualized as dynamic spheres around cached edits, with the radius determined by the distance between positive and negative samples. By adjusting the radius over time, BalancEdit allows for immediate edits, retains previous edits, and preserves correct model behav-

iors. Furthermore, since BalancEdit's codebooks do not alter model weights and are fully model-agnostic, they also pave the way for plug-and-play, cost-effective model editing. This is particularly useful for making critical spot-fixes between larger retraining efforts.

Our contributions are as follows: 1) We first formulate the generality-locality trade-off in multi-modal model editing and build a dataset named *OKEDIT* to empirically demonstrate it. 2) We introduce BalancEdit, an efficient method for multi-modal model editing that dynamically and effectively balances generality and locality without requiring training data beyond individual edits. 3) Our experiments reveal that BalancEdit outperforms baseline models and consistently achieves SOTA performance across a range of metrics.

## 2. Related Work

**Model Editing.** Model Editing, which has recently drawn a lot of attention, aims to make precise, targeted adjustments to the behavior of foundation models. This is crucial given that large foundation models may decay over time due to domain shifts and updates in knowledge, potentially leading to the dissemination of outdated factual information. Many approaches in this area suggest regularized-finetuning using auxiliary data, such as instances from the original training set or semantically-similar edits (Sinitsin et al., 2020), while obtaining this data is increasingly challenging. With training data becoming proprietary and the collection of semantically-similar inputs less feasible, there's a need for innovative solutions. Some recent strategies utilize meta-learning to forecast edits (Mitchell et al., 2022b;a; De Cao et al., 2021) or decompose weight updates into simpler components (Meng et al., 2022a;b). To make edits more targeted, techniques like MEND (Mitchell et al., 2022a) and ROME (Meng et al., 2022a) and GRACE (Hartvigsen et al., 2024) take cues from efficient finetuning strategies (Yu et al., 2023b; Huang et al., 2023; Yu et al., 2023a; Li et al., 2024; Tian et al., 2024). However, these methods sometimes demand additional finetuning and may overfit more

than traditional methods (Zhong et al., 2022) and few of them consider the locality property. MEND (Mitchell et al., 2021) notices the locality issue and designed a contrastive loss to keep the locality. Despite these advancements, there remains a substantial gap in model editing methods tailored for multi-modal models. Only limited research (Cheng et al., 2023) has explored the potential of multi-modal models in this context. In our work, we stick to this problem, investigating the trade-off between generality and locality in multi-modal model editing and offering an efficient method to address it.

**Large Vision Language Models.** Vision language models (Radford et al., 2021; Zhu et al., 2023; Li et al., 2023; 2022; Wang et al., 2024a; Zhou et al., 2024; Lin et al., 2024; Dai et al., 2024) are one of the key part in multi-modal learning, which aim to learn multi-modal foundation models with improved performance on vision language tasks (Antol et al., 2015; Guo et al.; Wang et al., 2024b). These models (Li et al., 2022; Liu et al., 2023a), by mapping image embeddings to text embedding space, are capable of interpreting image information and handling a wide array of tasks. They demonstrate impressive abilities in image understanding, generation, and reasoning. These capabilities, however, rely heavily on millions of high-quality training data (Schuhmann et al., 2022; 2021). Given that factual knowledge, especially visual information, changes over time, it is crucial to keep the model up-to-date. However, updating the model's behavior through retraining or fine-tuning is impractical due to exorbitant training costs. In this context, multi-modal model editing techniques, which allow for targeted edits, provide a feasible solution to this challenge.

# 3. Methods

## 3.1. Problem Formulation

The multi-modal model editing is to edit a multi-modal LLM $f_{base}$ that maps the image input ($i$) and text prompt ($t$) from the out-dated answer ($y^o$) to the new target prediction ($y^n$) with the updated model $f_{new}$. For the related inputs $R_{i,t}$, the updated model should give the target prediction, while for the unrelated inputs $U_{i,t}$, the prediction should be retained. In addition, when given a batch of inputs $(i, t, y^n) \in D_{\text{edit}}$, the updated model could remember all edits without forgetting previous edits. Specifically, the multi-modal model editing should follow the following properties: (1) **Reliability**. The updated model should output the target answers: $f_{new}(i, t) = y^n, (i, t, y^n) \in D_{\text{edit}}$; (2) **Generality**. The updated model should answer the target output given related inputs: $f_{new}(i', t') = y^n, (i', t') \in R_{i,t}$; (3) **Locality**. The updated model should keep the output retained on the unrelated inputs. $f_{new}(i', t') = f_{base}(i', t'), (i', t') \in U_{i,t}$. Thus, to achieve both generality and locality properties, it is necessary to distinguish the generality samples and local-

ity samples. Additionally, there are two *bonus properties*. (4) **Multiple Edits**. The model could edit multiple times without forgetting previous edits. (5) **Efficiency**. The model editing method should take minimal costs to edit a model, such as less training time and data costs.

## 3.2. BalancEdit

As illustrated in Fig. 2, to satisfy the aforementioned properties, we propose BalancEdit, an efficient model editing method for multi-modal models that dynamically determines the equilibrium between generality and locality without compromising the original model. BalancEdit operates by wrapping a selected layer of the pre-trained model with a BalancEdit module. This module consists of a codebook and a mechanism that dynamically determines the radius of the influence scope.

**BalancEdit Codebook.** To store the updated knowledge of the pre-trained multi-modal model, we design a discrete codebook at layer $l$ which contains three components.

- **Keys** ($K$): Each $k$ in the codebook is a representative embedding for a specific edit. Each $k$ stores the averaged embedding produced by the layer $l - 1$ for a specific question answer pair. Mathematically, it can be expressed as $K = \{k = \bar{h}_{i,t}^{l-1} | \bar{h}_{i,t}^{l-1} = \frac{1}{n} \sum f^{l-1}(i, t), \forall (i, t) \in D_{\text{edit}}\}$.

- **Transformations** ($V$): The transformation refers to a specific layer in the LLM. During an edit, we fine-tune this transformation layer to encode the new knowledge. Each transformation $v(\cdot)$ associated with a specific key $k$ stores the new weights with the updated knowledge. Typically, the transformation is fine-tuned with the model's finetuning loss with updated knowledge.

- **Influence radius** ($\mathcal{E}$): The radius $\epsilon$ corresponding to a key $k$ indicates the influence scope of a $(i, t, y^n)$ pair. It serves as a threshold for similarity matching. The edited transformation is activated only if the embedding falls within the influence radius. The radius varies for each key, and is determined by the positive and negative samples of a specific knowledge pair $(i, t, y^n)$.

**Codebook Constructions.** To make an edit, the BalancEdit module needs to create a new codebook entry $(\bar{h}_{i,t}^{l-1}, v(\cdot), \epsilon)$. The key is the averaged embedding generated by the layer $l - 1$, which is an anchor point for lookup. Thus, when a new question is passed into $f$, the codebook is activated to compare whether the embedding relates to any key in the codebook. If the embedding falls within the influence scope of a key, the edited transformation is activated to generate a new embedding for layer $l + 1$; otherwise, the original transformation is retained to process the question.

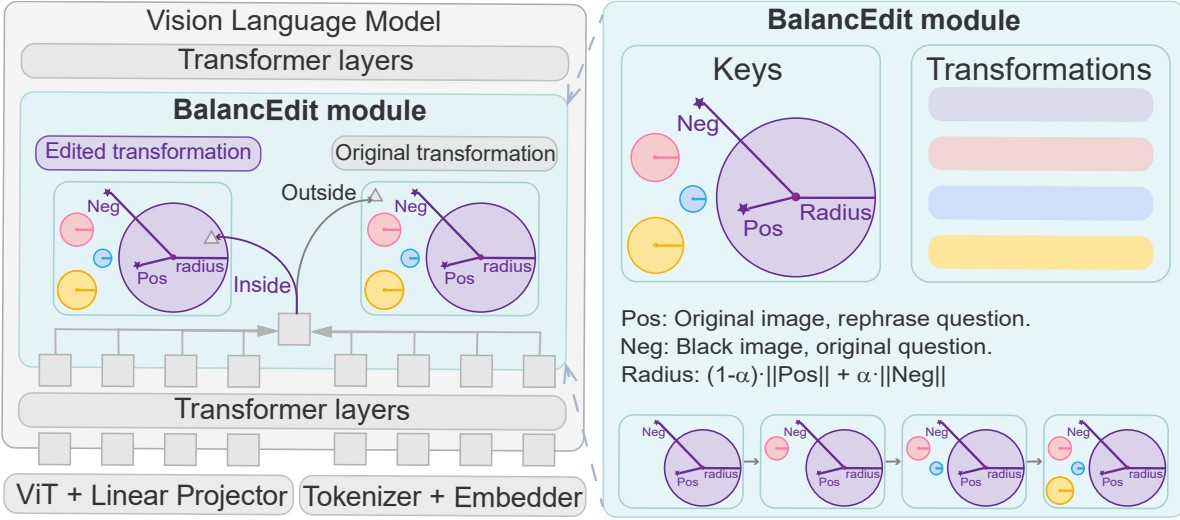

*Figure 2.* Overview of our BalancEdit framework. BalancEdit makes edits by learning, saving, and retrieving transformational edits between layers. The BalancEdit module consists of discrete keys, transformations, and a dynamic influence radius. Additionally, the BalancEdit module can handle multiple edits over time by adding new entries to the module.

The formulation is as follows:

$$h_{i,t}^l = \begin{cases} v_k(h_{i,t}^{l-1}), & \text{if } \min(d(h_{i,t}^{l-1}, K)) \le \epsilon_k \\ f^l(h_{i,t}^{l-1}), & \text{otherwise} \end{cases} \quad (1)$$

**Editing Transformations.** When a new fact requires an update, the transformation is revised to incorporate this new fact and knowledge. To ensure that the transformation accurately learns the new fact, we finetune the transformation layer directly using backpropagation through the language learning loss. The target transformation $v^*$ can be formulated as:

$$v^* = \arg\min_v \boldsymbol{L}(f_{new}(i, t), y^n), \quad (2)$$

where $\boldsymbol{L}$ is the language model loss, specifically the next-token prediction loss used by the base LLM. This loss ensures that the updated model $f_{\text{new}}$, when processing input $(i, t)$, generates the desired new answer $y_n$. Specifically, if the key is empty or the new fact falls outside the influence scope of existing keys, the transformation is directly finetuned from the original transformation layer. However, there may be instances where the new fact overlaps with the existing keys. In such cases, we finetune the transformation layer from the previously edited transformation to prevent catastrophic forgetting. Additionally, if the new key directly conflicts with previous edits, we will discard the previous entry and add a new one to update the knowledge. To ensure the universality, we primarily utilize the basic full fine-tuning approach as the transformation method. This involves adjusting the weights of the neural network to better align with the newly introduced or modified knowledge

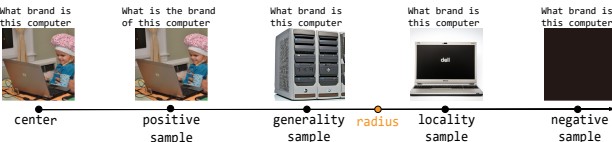

*Figure 3.* Illustration of influence radius determination

without altering the overall architecture of the model. The parameters that are tuned include all the weights within the specific layer of the network.

**Influence Radius Determination.** As shown in Fig. 1, each fact has its unique influence scope. However, existing methods do not consider the dynamic influence scope during the editing process, which results in an imbalanced generality-locality trade-off, as illustrated in Table 1. To address this issue, BalancEdit incorporates a dynamic influence radius determination mechanism. As depicted in Fig. 3, the knowledge of the fact is at the center of the influence scope. Ideally, the radius should encompass the majority of generality samples, while excluding locality samples. Since similar semantic sentences will result in close embeddings (Liu et al., 2023b; Menon & Vondrick, 2023), we can use it to find an efficient way to approximate this process. Specifically, we construct positive and negative samples to dynamically estimate the influence scope **without** model training or external knowledge.

To construct a positive sample, we need to design a general rephrasing method that is highly similar to the fact itself. We find that rephrasing the text will not affect the semantic in-

formation of the edited knowledge. Therefore, we rephrase the text prompt $t$ while keeping the image input $i$ unchanged. The positive sample can be formulated as $(i, R(t))$, where $R(t)$ denotes the rephrased text prompts. The generation of a rephrased prompt is efficient, requires no additional data or training process, and can be generated directly by the backbone model.

On the other hand, the negative sample should be close to the border of locality samples to accurately estimate the radius. Additionally, the generation process should be efficient and fact-agnostic. In this case, we use a pure black image as the image input, which contains no semantic information on the image side. The choice of black images as a proxy for out-of-scope knowledge is based on their characteristic as minimal or null visual signals. This makes them universally applicable negative samples across various visual recognition tasks. Furthermore, the generation of a negative sample is highly efficient, and can be applied to almost all knowledge editing tasks.

After obtaining the positive and negative samples, we can estimate the influence radius by aggregating the distances between the center and the constructed samples. Specifically, the radius could be formulated as:

$$\epsilon = (1 - \alpha) \cdot d(Pos, k) + \alpha \cdot d(Neg, k), \qquad (3)$$

where $\alpha$ is the hyperparameter to adjust the distance, $d(\cdot)$ denotes the distance function, and $k$ is the key in the codebook entry which also represents the center of the influence scope.

# 4. Experiments

To evaluate the properties discussed in Sec. 3.1, we conduct experiments from three perspectives: 1) The primary motivation of BalancEdit is to balance generality and locality. Therefore, we create a dataset named OKEDIT to address the quality issues of existing datasets and conduct experiments on it. 2) We assess the performance of multiple edits, and 3) we compare the training time and the data costs of an editing method to evaluate its efficiency.

## 4.1. Datasets and Backbone Models

**Datasets.** Since there are few published vision language model editing datasets, we perform extensive experiments on two such datasets in the vision question answering task (Antol et al., 2015): 1) **MMEDIT**(Cheng et al., 2023), the first multi-modal model editing dataset based on the VQA-v2(Goyal et al., 2017) dataset, which includes 2093 testing samples; However, this dataset has its limitations as shown in table 2. The content of images generated from image caption prompts can deviate from the original images, leading to inconsistencies and potentially less accurate eval-

| | # Train | # Test | Generality | Locality | Goal |
|---|---|---|---|---|---|
| MMEDIT | 6036 | 2093 | 1 per question | random sample, easier eval | visual understanding |
| OKEDIT | 9009 | 5046 | 10 per question | semantic sample, harder eval | visual reasoning with open question |

*Table 2.* Statistics comparison between MMEDIT and our OKEDIT.

uations. 2) We introduce a new dataset, **OKEDIT**, based on the OKVQA dataset (Marino et al., 2019), which includes 5046 testing samples, encompassing **over 20 unique categories** such as vehicles, people, plants, animals, geography, history, language, brands, science and technology. Unlike MMEDIT, OKEDIT enhances the quality of the rephrased images and adjusts the difficulty of the locality samples to evaluate the trade-off between generality and locality. Detailed information about the datasets is provided in Appendix A.

**Backbone Models.** Following previous work (Zheng et al., 2023), we adopt two vision language models as the base models. **MiniGPT-4** (Zhu et al., 2023) is a powerful vision language model, leveraging Vicunna (Chiang et al., 2023) as the language model and a Vit-G/14 from EVA-CLIP (Sun et al., 2023) and a Q-former as the image encoder. **BLIP-2 OPT** (Li et al., 2023) utilizes a lightweight Q-former to bridge the gap between vision modality and text modality, where the ViT-L is adopted in the vision block, and the unsupervised-trained OPT model (Zhang et al., 2022) is used for decoder-based LLM.

**Metrics.** Following previous work (Zheng et al., 2023), we adopt the Editing Success Accuracy (**Acc**); Text Generality (**T-Gen**); Image Generality (**I-Gen**); and Locality (**Loc**) as the main metrics. To quantify the trade-off between generality and locality, we introduce the harmonic mean (**HM**) of the T-Gen, I-Gen and Loc. The detailed informations are in Appendix C.

## 4.2. Baselines

We compare four model editing methods with different mechanisms. First, finetuning (**FT**) is a basic model editing method. To ensure a fair comparison, we only fine-tune the specific layer of the pre-trained model, maintaining the same parameter sizes. Second, In-context Knowledge Editing (**IKE**) is an in-context learning model editing method originally designed for pure language models. We have revised the method to adapt it to vision-language models. It utilizes an unsupervised retriever to prompt relevant facts from the training set. Additionally, **MEND**(Mitchell et al., 2021), a metalearning-based model editing method, requires extensive in-distribution training data to learn a meta-network that predicts the edited weights of the pre-trained model. Finally, we adapt **GRACE**(Hartvigsen et al., 2024) to vision language models. GRACE, a memory-augmented model edit-

| Dataset | Method | Pretrain | Backbone | | | | | | | | | |
| | | | miniGPT4 | | | | | BLIP-2 OPT | | | | |
| | | | Acc↑ | T-Gen↑ | I-Gen↑ | Loc↑ | HM↑ | Acc↑ | T-Gen↑ | I-Gen↑ | Loc↑ | HM↑ |
| MMEDIT | Base | ✗ | 15.04 | 14.21 | 13.56 | NA | NA | 8.50 | 8.52 | 6.89 | NA | NA |
| | FT | ✗ | 96.53 | 95.88 | 96.20 | 3.20 | 9.00 | 99.96 | 99.41 | 97.05 | 0.27 | 0.80 |
| | IKE | ✓ | 100.00 | 95.57 | **100.00** | 15.47 | 20.07 | 99.83 | 94.47 | 99.58 | 11.96 | 28.77 |
| | MEND | ✓ | 98.39 | 96.58 | 97.77 | 68.82 | 85.43 | 97.23 | 95.86 | **96.81** | 69.40 | 85.29 |
| | GRACE | ✗ | 79.82 | 74.49 | 70.11 | **91.66** | 77.72 | 74.27 | 62.90 | 35.24 | **90.26** | 54.19 |
| | BalancEdit (Ours) | ✗ | **100.00** | **99.90** | 98.91 | 71.74 | **88.08** | **100.00** | **99.16** | 90.30 | 80.04 | **89.14** |
| OKEDIT | Base | ✗ | 30.42 | 45.40 | 72.21 | NA | NA | 14.35 | 13.96 | 15.22 | NA | NA |
| | FT | ✗ | 99.69 | 99.45 | 99.38 | 5.52 | 14.90 | 99.97 | 99.54 | 96.77 | 0.43 | 1.27 |
| | IKE | ✓ | 99.71 | 97.78 | **99.76** | 17.45 | 38.68 | 99.35 | 94.20 | **99.66** | 13.29 | 31.28 |
| | MEND | ✓ | 94.44 | 90.80 | 95.39 | 36.20 | 61.07 | 90.82 | 82.82 | 88.25 | 28.89 | 51.70 |
| | GRACE | ✗ | 87.84 | 28.31 | 29.46 | **99.99** | 37.84 | 54.13 | 50.67 | 28.30 | **94.48** | 45.69 |
| | BalancEdit (Ours) | ✗ | **100.00** | **99.87** | 76.46 | 53.14 | **71.58** | **100.00** | **98.89** | 65.38 | 61.18 | **71.85** |

*Table 3.* Comparison results of BalancEdit with the model editing baselines on two backbone models. Base refers to the backbone model without any knowledge editing. The pretrain column indicates whether a model editing method requires pre-training model or the training data. The best results are shown in **Bold**.

ing method, also supports lifelong model editing. It caches the target value of the updated fact, achieving lightweight model editing.

### 4.3. Implementation Detail

In our comparisons of Finetuning, MEND and GRACE, we explore learning rates of 1.0, $1e^{-1}$, $1e^{-2}$, $1e^{-3}$, $1e^{-4}$, and $1e^{-5}$. We observe that Finetuning, Memory, and MEND perform best with $1e^{-2}$. The choice of layer to edit is another hyperparameter for all editors. In all our editor comparisons, each editor modifies the same layer. For miniGPT-4, this is the dense layer of the llama block (`llama_model.model.layers[31].mlp.up_proj`), for BLIP-2 OPT moder, it is the OPT decoder layer (`opt_model.model.decoder.layers[31].fc2`). Recent work supporting the importance of selecting the correct layers to fine-tune corroborates this (Cheng et al., 2023). However, it's important to note that the choice of layer is a practical hyperparameter: for comparison purposes, we ensure editors are compared when editing the same layers. For the distance function, we use the Euclidean distance if it is not explicitly mentioned. We select $\alpha$ using a small held-out set of only 5 unrelated samples. Since different models may have different latent feature distributions, $\alpha$ is treated as model-dependent. Once chosen, the same $\alpha$ is fixed per model (e.g., $\alpha$=0.2 for MiniGPT-4) for all evaluations.

### 4.4. Comparisons to Existing Methods

Table 3 presents the main results of our BalancEdit and other baseline methods on the VQA task. We observe that our BalancEdit significantly outperforms the existing editing methods without requiring additional training data. Specif-

ically, we examine both the accuracy and the trade-off between generality and locality. First, in terms of editing success accuracy, BalancEdit achieves the highest performance, resulting in 100% editing success across all datasets and backbones. In contrast, baseline models do not consistently reach this level of performance. This demonstrates that our BalancEdit satisfies the **Reliability Property**.

For the Generality metric, BalancEdit achieves the best text generality performance compared to other methods. For instance, BalancEdit shows a 70% improvement in text generality accuracy over the GRACE method. Additionally, it reaches comparable performance in image generality. Since the MMEDIT dataset is relatively simple, the performance is very similar across all model editing methods, converging around 99%. However, in the challenging OKEDIT dataset, where we focus on balancing trade-off performance, we must compromise on the more difficult aspects of image generality. The high generality performance underscores the **Generality Property** of our method.

For locality performance, BalancEdit consistently achieves the best results, with the exception of the GRACE method, which primarily focuses on specific local edits. Specifically, BalancEdit shows an improvement in locality 20% to 80% compared to other baseline methods. For example, on the OKEDIT dataset using the BLIP-2 OPT backbone, BalancEdit outperforms the MEND method by 30%, despite the fact that MEND requires extensive training data and time. This further validates the **Locality Property** of our BalancEdit method.

To compare the overall performance in balancing the trade-off between locality and generality, we calculate the harmonic mean of T-Gen, I-Gen, and Loc. Our BalancEdit method achieves the highest scores compared to other base-

| | Sequential | Acc↑ | T-Gen↑ | I-Gen↑ | Loc↑ | HM↑ |
|---|---|---|---|---|---|---|
| FT | ✗ | 99.25 | 99.21 | 98.64 | 0.74 | 2.18 |
| IKE | ✗ | 100.00 | 96.86 | 100.00 | 16.91 | 37.75 |
| MEND | ✗ | 93.74 | 89.98 | 37.49 | 37.49 | 62.14 |
| GRACE | ✗ | 87.78 | 25.96 | 24.21 | 99.99 | 33.39 |
| BalancEdit (Ours) | ✗ | 100.00 | 100.00 | 72.31 | 54.40 | **71.07** |
| BalancEdit (Ours) | ✓ | 100.00 | 99.70 | 72.29 | 46.25 | 65.95 |

*Table 4.* Comparison results of BalancEdit with the model editing baselines about multiple sequential editing. The sequential column indicates whether the method uses sequential editing or not.

lines across all experimental combinations, demonstrating the minimal trade-off between generality and locality performance. Specifically, in the simpler MMEDIT dataset, BalancEdit outperforms the strongest baseline, MEND, by 3% and surpasses other baselines by up to 89%. Furthermore, in the more challenging OKEDIT dataset, our results are even more impressive, outperforming the MEND baseline by between 10% and 20%. As expected, these performances highlight the effectiveness of our dynamic influence scope mechanism and validate the **Reliability, Generality**, and **Locality Properties** of our method.

### 4.5. Sequential Editing Evaluation

To further investigate performance across multiple sequential edits, we evaluated our BalancEdit system on 50 sequential edits using the OKVQA dataset with a miniGPT-4 backbone. The results, as shown in Table 4, indicate a slight drop in performance for sequential edits compared to non-sequential ones. Specifically, metrics such as edit success accuracy and generality remain comparable with those observed in non-sequential editing scenarios, suggesting that the system's reliability and generality are maintained. Although there is a slight decrease in locality performance, it still exceeds that of other baselines. This decrease is expected, as an increase in the number of keys can lead to unwanted collisions, potentially degrading performance. Notably, despite the slight performance reduction in sequential editing, our BalancEdit system continues to outperform baseline models that do not incorporate sequential edits. This performance across multiple edits substantiates the **Multiple Edits Property** of our system.

### 4.6. Efficiency Evaluation

We compare the efficiency of our editing approach with recent advanced baselines, focusing on both time and data efficiency. Time efficiency encompasses both training and editing time, while data efficiency refers to the amount of additional data required for editing.

**Time Efficiency.** Training time is divided into two components, as detailed in Table 5. The first component is pre-training time, which involves either pre-training the model editing method or preparing the augmented index, such as

| | Training time (h) | Editing time (s) |
|---|---|---|
| FT | 0 | 3.91 |
| IKE | 12 | 0.38 |
| MEND | 22 | 1.48 |
| GRACE | 0 | 32.67 |
| BalancEdit | 0 | 8.04 |

*Table 5.* Time efficiency evaluation results on BLIP-2 OPT.

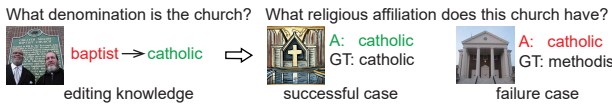

*Figure 4.* An example of the interpretable output

training a meta-net. For instance, MEND, a meta-learning method, requires 22 hours to pre-train on 6,346 training samples. IKE, a retrieval-augmented in-context learning method, needs 12 hours to index 6,346 knowledge facts in advance.

On the other hand, editing time refers to the duration required to edit a single new fact. We compare with GRACE as both methods are types of memory-augmented model editing. A successful edit with our method takes approximately 8.04 seconds, whereas GRACE takes 32.67 seconds, making our editing speed three times faster than GRACE. IKE requires less editing time because it bypasses training and instead retrieves the most similar fact.

**Data Efficiency.** Similar to training time costs, the requirement for additional training data significantly influences the feasibility of model editing methods. Our BalancEdit does not require any extra data, as it can generate both positive and negative samples internally. Specifically, a rephrased question for a positive sample can be obtained by querying the backbone model, and a black image for a negative sample can be directly generated. This efficiency supports the Efficiency Property. In contrast, methods like MEND and IKE require extensive additional in-distribution data, leading to less feasibility for real-world scenarios.

### 4.7. Interpretability

**Interpretable Codebook.** The codebook is interpretable because the editing knowledge is explicitly stored, with each entry corresponding to an update in knowledge and its specific influence scope. Additionally, the codebook is detachable and can be thoroughly inspected, allowing edits to be easily located and detected. Each piece of updated knowledge has an entry in the codebook, enabling it to be reversed without impacting the model, particularly in sequential editing scenarios. This interpretable codebook minimizes harm to the model while maintaining controllability.

**Interpretable Inference.** Existing model editing methods

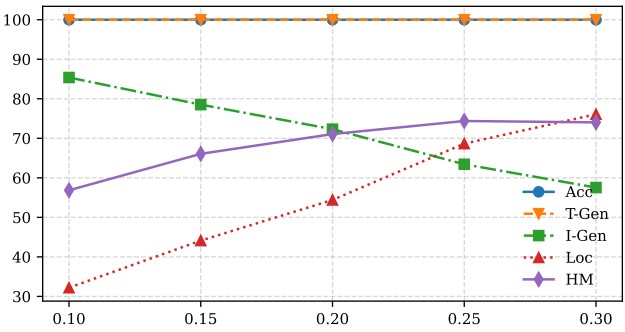

*Figure 5.* Results of the effect of the hyperparameter $\alpha$.

| Dataset | Function | Acc↑ | T-Gen↑ | I-Gen↑ | Loc↑ | HM↑ |
|---|---|---|---|---|---|---|
| MMEDIT | Euc | 100 | 99.9 | 98.91 | 71.74 | 88.08 |
| | Cos | 100 | 99.9 | 97.96 | 76.28 | 90.01 |
| OKEDIT | Euc | 100 | 99.87 | 76.46 | 53.14 | 71.58 |
| | Cos | 100 | 99.87 | 84.26 | 42.37 | 65.95 |

*Table 6.* Results of the effect of different distance function.

| Negative Anchor | Acc↑ | T-Gen↑ | I-Gen↑ | Loc↑ | HM↑ |
|---|---|---|---|---|---|
| Black | 100.00 | 99.00 | 69.16 | 59.99 | 72.76 |
| White | 100.00 | 99.00 | 65.79 | 63.85 | 73.23 |

*Table 7.* Comparative results using alternative negative anchors

typically update the knowledge within the model but do not provide a means to trace how these updates influence the model's output. Specifically, while the model's outputs may change, it is unclear how these changes are influenced by the updates and whether they are relevant to the posed question. In contrast, BalancEdit offers a human-understandable explanation for adjusting model behavior. As illustrated in Figure 4, we edit the counterfactual scenario where 'baptist church' is changed to 'catholic church'. In a successful case, we correctly answer the question because the image displays a symbol of the baptist church, even though it is not explicitly shown. According to the closest BalancEdit key, we can infer that the output is influenced by the edited knowledge. In contrast, in a failure case, we can determine that the incorrect prediction arises because the image closely resembles the edited fact.

### 4.8. Ablation Study

In this ablation study, we analyze the key design choices of BalancEdit. We first examine the effect of the hyperparameter $\alpha$, which controls the trade-off between generality and locality. We then evaluate the impact of alternative negative anchors and different distance functions to validate the generalizability. Additionally, we compare different negative sampling approaches and editing layers to assess the robustness of our method across different configurations. Finally, we conduct an extreme case analysis to demonstrate the behavior of BalancEdit in challenging scenarios.

**Effect of the Hyperparameter.** In this study, we conducte a series of experiments on a subset of the OKVQA dataset to investigate how the parameter $\alpha$ affects the trade-off between generality and locality in model editing. As illustrated in Figure 5, we vary $\alpha$ from 0.1 and 0.3. The results, depicted in the figure, show that the editing success accuracy and text generality metrics consistently maintain a 100% accuracy rate. This stability is attributed to these metrics being closely tied to the key, with changes in the radius having no significant impact on them. However, the image generality metric, which is more challenging, shows a decline as $\alpha$ increases. This trend is anticipated because questions

related to image generality tend to deviate from the key, despite sharing similar semantic content. Consequently, as the radius decreases, the edited model tends to overlook these questions. Conversely, the model's performance on locality improves with an increase in $\alpha$. A smaller radius helps preserve the integrity of unrelated questions, ensuring that their answers remain unchanged. In this scenario, we observe that the harmonic mean of generality and locality initially increases and then decreases, further validating the existence of this trade-off. However, our method continues to achieve relatively high performance.

**Effect of Alternative Negative Anchors.** To further validate the effectiveness of our approach, we conducte experiments using various negative anchors, including white negative images, on a subset of the OKEDIT dataset. As shown in Table 7, both black and white negative samples achieve 100% editing accuracy and exhibited a high harmonic mean in the locality-generality trade-off. The performance metrics for both white and black negative anchors, such as accuracy, generalization metrics, and locality, are remarkably consistent. The slight variations in the locality and I-Gen metrics suggest that white images can function as effective negative anchors, which also lack significant discriminative information. This consistency across different negative anchors highlights the robustness and adaptability of our pipeline in various settings and confirms our assumptions.

**Effect of the Distance Function.** The distance function serves as a method for calculating the similarity between two embeddings. In particular, we employ the Euclidean distance (Euc) and cosine similarity (Cos) as the distance metrics. To assess the versatility of our BalancEdit in terms of the distance function, we compare these two popular distance functions, as illustrated in Table 6. We find that the results between them are remarkably similar. Specifically, both achieve around 100% editing success accuracy and text generality. While there are some differences in image generality and locality, both functions yield comparable results. This is expected as the distance function alters the similarity between embeddings, but the semantic meanings for the positive and negative samples are still preserved.

| Method | Edit Acc | T-Generality | I-Generality | Locality | HM | Model |
|---|---|---|---|---|---|---|
| BalancEdit | 100.00 | 98.89 | 65.38 | 61.18 | **71.85** | BLIP-2 OPT |
| Random Negative Sample | 100.00 | 100.00 | 49.12 | 65.08 | 65.61 | BLIP-2 OPT |
| BalancEdit | 100.00 | 99.87 | 76.46 | 53.14 | **71.58** | MiniGPT-4 |
| Random Negative Sample | 100.00 | 99.00 | 66.93 | 45.92 | 64.08 | MiniGPT-4 |

*Table 8.* Ablation study on negative sampling approaches

| Layer | Acc | T-Gen | I-Gen | Loc | HM | Model |
|---|---|---|---|---|---|---|
| 30 | 100.00 | 100.00 | 64.03 | 66.39 | 73.75 | MiniGPT4 |
| 31 | 100.00 | 99.87 | 76.46 | 53.14 | 71.58 | MiniGPT4 |
| 30 | 100.00 | 100.00 | 78.43 | 44.99 | 66.70 | BLIP-2 OPT |
| 31 | 100.00 | 98.89 | 65.38 | 61.18 | 71.85 | BLIP-2 OPT |

*Table 9.* Ablation study on different editing layers.

| $\alpha$ | Acc | T-Gen | I-Gen | Loc | HM |
|---|---|---|---|---|---|
| 0 | 100.00 | 100.00 | 95.19 | 16.30 | 36.65 |
| 1 | 100.00 | 45.94 | 24.22 | 100.00 | 41.06 |

*Table 10.* Results of extreme cases of $\alpha$

This success highlights the effectiveness of our codebook strategy, as it can dynamically adapt to different distance functions while maintaining a similar influence scope.

**Effect of Negative Sampling Approaches.** To verify our negative samples constuction approach, we compare our method with random negative sampling approach by randomly choosing irrelevant text-pair samples. As shown in Table. 8, BalancEdit consistently outperforms the random baseline in harmonic mean, demonstrating a better balance between generality and locality. This supports the effectiveness of our black image-based negative anchor, which offers a fact-agnostic, consistent, and efficient way to define a lower bound in the representation space. In contrast, random negative samples rely on external, unrelated examples and are often unstable in both quality and relevance, requiring assumptions that may not hold across diverse domains. Our method avoids these issues, making it more robust and scalable for real-world editing scenarios.

**Effect of different editing layers.** To evaluate the robustness of our method across different editing layers, we apply our method to two separate layers, as shown in Table 9. The results show that our method maintains strong performance even when a different layer is chosen. Specifically, we observe higher harmonic mean performance on the MiniGPT-4 model, further demonstrating the robustness of our approach.

**Extreme case analysation.** In order to study the extreme case of hyperparameter, we conduct ablation experiments using extreme values of $\alpha$ on MiniGPT4 with OKEDIT dataset in Table 10. When $\alpha = 0$, the influence radius is entirely determined by the negative sample, which leads to over-generalization and reduced locality, e.g., 16.30 lo-

cality score, as the edit is applied too broadly across the representation space. In contrast, when $\alpha = 1$, the radius is determined solely by the positive sample, resulting in over-localization and poor generalization, e.g., 24.22 image generality score, since the edit is confined to a region that is too narrow. These results highlight the importance of balancing positive and negative influences to achieve both generality and locality.

## 5. Limitations

Multi-modal model editing is a novel and challenging field, with the balance between generality and locality remaining largely underexplored. It is evident that employing similarity search across a model's layers inevitably slows down inference times (Hartvigsen et al., 2024), despite reducing the need for extensive training. Thus, accelerating inference time represents a crucial area for future improvements.

Another limitation is the memory-augmented approach's handling of multi-hop model editing. The key-based similarity search struggles to capture multi-hop queries that depend on newly introduced knowledge, often due to the ambiguity of real-world facts. For example, if the CEO of X (formerly Twitter) were to change to Elon Musk, it would be difficult to update the response to the question, 'Which social app is headed by the leader of SpaceX?' A potential solution to this problem could involve dynamically defining the fine-grained influence scope, which would allow for more precise adjustments to changes in real-world facts and their implications for multi-hop questions.

## 6. Conclusion

In conclusion, we identified the limitation of existing imbanlanced generality and locality in model editing. Specifically, we formulated the generality-locality trade-off, and developed a specialized dataset, *OKEDIT*, to empirically explore this phenomenon. In addition, we introduced BalancEdit, an innovative approach for multi-modal model editing that efficiently balances the generality and locality of edits. Our method reduces the need for extensive retraining or fine-tuning, relying solely on the data provided by individual edits. The experimental results demonstrate that BalancEdit significantly outperforms existing baseline models, consistently achieving state-of-the-art performance.

## Acknowledgments

The work is supported in part by the U.S. Office of Naval Research Award under Grant Number N00014-24-1-2668, and the National Science Foundation under Grants IIS-2316306 and CNS-2330215.

## Impact Statement

This paper presents work whose goal is to advance the field of Machine Learning. There are many potential societal consequences of our work, none of which we feel must be specifically highlighted here.

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

# A. Dataset

Although numerous studies have been conducted on knowledge editing in Large Language Models (LLMs), research in the context of Large Vision-Language Models (LVLMs) remains relatively sparse. Only one benchmark, MMEDIT (Cheng et al., 2023), has delved into this domain within LVLMs. This benchmark extend the concepts of Reliability, Generality, and Locality from LLM editing, incorporating diffusion-model-generated images in its Generality evaluation.

However, this dataset has its limitations as shown in table 2. The content of images generated from image caption prompts can deviate from the original images, leading to inconsistencies and potentially less accurate evaluations. Furthermore, the scarcity of data in the only existing benchmark presents a significant harm the progress in LVLM knowledge editing. Therefore, the availability of more data would greatly aid in the development and refinement of techniques in this field.

In our research, we utilize the multimodal VQA dataset OKVQA (Marino et al., 2019), which provides hard image questions with difficult visual reasoning and open knowledge. Furthermore, the OKVQA dataset provide detailed question categories which could be used to evaluate the editing method on different question types.

## A.1. Dataset Construction Details

OKEDIT dataset are constructed to provide pairs of edit input $(i, t)$ and a counterfact answer $y^n$. The edit label is not necessarily the 'correct' label; the goal is to provide realistic instances of the types of data we would expect to see during test. For example, given the $i$ as *a HP brand computer*, and $t$ = *What is the brand of it*, and $y^e$ is the *lenovo*, even though it never happens currently. However, this fictitious example is still a useful assessment of our model's ability to perform the general type of edit of 'change a name of an item'.

To evaluate the **text generality**, we generate some samples using the rephrasing methods. Specifically, we use the GPT-4 API to generate the rephrased questions, with the following command.

*"Please rephrase the following question in {num_versions} different ways: {question}."* where we generate 10 rephrased questions.

For the **image generality**, we need to generate semantic similar images. To get the semantic meaning of a specific image in the question context, we first question the GPT-4 which objects and scene should be in the image.

*"Given (question: {question}, answer: {answer}), what object should be in the image? Short answer. The objects in*

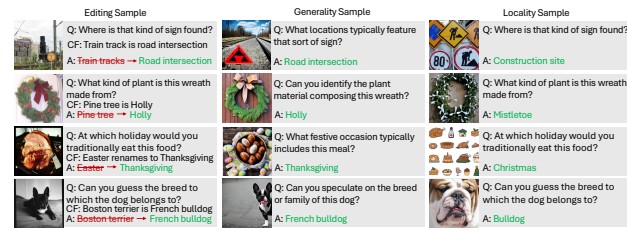

*Figure 6.* Examples of our OKEDIT dataset. 'CF' represents the edited counterfactual knowledge. The red color indicates the outdated answer and the green color indicates the updated correct answer.

*the image should be "*

After we obtain the image object, we can ask the diffusion model to generate it with the image object. For each image, we also generate 10 images for evaluation.

For the **locality** evaluation, we try to generate an image that is similar enough the original image but it still unrelated to it. To achieve that, we have three steps generation. Fisrt, we will determine the locality answer with high similarity with the target answer, with the help of GPT-4.

*"Given (question: {question}, A: [{answer}], B: [{counterfact_answer}]), what could be another option? Short answer. C: []"*

Then, we can follow the same steps as in the image rephrasing process to generate locality images, including obtaining image objects and generating images with diffusion model.

# B. Dataset Samples

We present several examples from our OKEDIT dataset in Figure 6. Our dataset offers high-quality images and samples of counterfactual knowledge editing. Additionally, some samples incorporate common sense knowledge, which adds complexity to the editing tasks. These characteristics enhance the overall quality of our dataset in comparison to the existing MMEDIT dataset.

# C. Metrics

(1) **Reliability**. The updated model should output the target answers: $f_{new}(i, t) = y^n, (i, t, y^n) \in D_{edit}$; (2) **Generality**. The updated model should answer the target output given related inputs: $f_{new}(i', t') = y^n, (i', t') \in R_{i,t}$; (3) **Locality**. The updated model should keep the output retained on the unrelated inputs. $f_{new}(i', t') = f_{base}(i', t'), (i', t') \in U_{i,t}$. Additionally, there are two *bonus properties*. (4) **Multiple Edits**. The model could edit multiple times without forgetting previous edits. (5) **Efficiency**. The model editing method should take minimal costs to edit a model, such as

less training time and data costs.

**Reliability** The updated model should output the target answers correctly.

$$\mathbb{M}_{\text{reliability}} = \underset{(i,x,y^n)\in D_{\text{edit}}}{\mathbb{E}} \mathbb{1}\{f_{new}(i,t) = y^n\} \quad (4)$$

**Text Generality** The updated model should answer the correct answer given the related rephrased question.

$$\mathbb{M}_{\text{T-Gen}} = \underset{(i,t,y^n)\in D_{\text{edit}}}{\mathbb{E}} \mathbb{1}\{f_{new}(i,R(t)) = y^n\} \quad (5)$$

**Image Generality** Similarly, the updated model should answer the correct answer given the similar images.

$$\mathbb{M}_{\text{I-Gen}} = \underset{(i,t,y^n)\in D_{\text{edit}}}{\mathbb{E}} \mathbb{1}\{f_{new}(R(i),t) = y^n\} \quad (6)$$

**Locality** The updated model should not change the irrelevant knowledge that is stored in the original model.

$$\mathbb{M}_{\text{Loc}} = \underset{(i',t',y^n)\in U_{i,t}}{\mathbb{E}} \mathbb{1}\{f_{new}(i',t') = f_{base}(i',t')\} \quad (7)$$

## D. Theoratical Analyse

Here is a brief theoretical proof about the effectiveness of our radius.

**Lemma**: Embeddings of semantically similar concepts are close in the embedding space.

**Proof.** 1. *Definition of Embeddings*: Embeddings are vector representations of concepts in a high-dimensional space. Formally, let $f : C \to \mathbb{R}^d$ be an embedding function that maps a concept $c \in C$ to a vector $f(c) \in \mathbb{R}^d$.

2. *Semantic Similarity*: Semantic similarity between two concepts $c_1$ and $c_2$ can be quantified using a similarity measure $S(c_1, c_2)$. Common choices include cosine similarity, Euclidean distance, or dot product.

3. *Objective of Embedding Training*: During the training of embeddings, the objective is typically to maximize the similarity of embeddings for semantically similar concepts and minimize it for dissimilar ones.

$$S(f(c_1), f(c)) < S(f(c_2), f(c)), \text{ if } S(c_1, c) < S(c_2, c) \quad (8)$$

**Assumption**: The generality sample (G) is semantically more similar to the editing knowledge (E) than the locality sample (L). That is, $S(G, E) < S(L, E)$. According to Lemma 1, we can state that $S(f(G), f(E)) < S(f(L), f(E))$.

**Conclusion**: In this case, we can find a radius $\epsilon$ such that

$$S(f(G), f(E)) < \epsilon < S(f(L), f(E)), \quad (9)$$

where

$$\epsilon = \alpha \cdot S(f(G), f(E)) + (1 - \alpha) \cdot S(f(L), f(E)). \quad (10)$$

## E. Baselines

**Finetune** In this method, we carry out a fine-tuning process on a selected layer of the pretrained model using Adam optimization for a fair comparison, while keeping all other layers fixed. For the training loss, the Cross Entropy loss is used for fine-tuning.

**IKE (Zheng et al., 2023)** IKE (In-Context Knowledge Editing introduces a system that utilises an unsupervised retriever. This retriever uses cosine similarity to pinpoint pertinent demonstrations from the training set. This method is grounded in the principles set forth by (Liu et al., 2022) and aims to insert new factual knowledge into language models in a non-disruptive fashion, eliminating the need for direct parameter updates. IKE's approach ranks demonstrations according to their resemblance to the editing target and organizes them in sequence to form a supplementary knowledge base that steers the model's generation process. This technique not only conserves the model's existing knowledge base but also presents a scalable and efficient method to refresh factual information. It shows considerable promise in mitigating unintended side effects, such as over-editing and knowledge forgetting, typically linked with gradient-based editing methods. However, it is also designed for pure text models for retrievel, to make it adapt to vision language models, we used composed embedding as the augmented database, such that it can retrieve the image information as well.

**MEND (Mitchell et al., 2022a)** MEND employs a hypernetwork to predict new weights for a selected layer of a pre-trained model by estimating the low-rank decomposition of the weight matrix of the layer. The hypernetwork is trained on a set of training edits, which comprises a new edit, a set of inputs that are semantically equivalent to the edit, and samples from the model's pre-training data. However, MEND is designed for the language model, to fit it to the vision language model, we keep the vision encoder fixed and only choose the language model layer for finetuning. In addition, in our situation, we only have single edits that are streaming in, we train the hypernetwork to predict updated weights as edits stream in using continuous fine-tuning.

**GRACE (Meng et al., 2022a)** GRACE is a lifelong model editing method for large language models. It handles sequential edits with a discrete key-value codebook. GRACE replace one layer to a GRACE adaptor which stores the key-value pair of the target edits, where the key is the last embedding of the key for the text prompt and value is trained

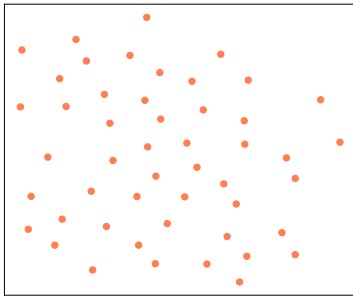

*Figure 7.* T-sne figure of key distribution in sequential editing.

by backpropagation with the target results. Keep handling the key conflicts could make it successfully deal with the multiple sequential editing in language models. However, to adapt it to the vision language model, we select the language part as the edited layer and prepend the image embedding before the text prompt so that it can be regarded as long text questions.

## F. Training Specifications

We use the Adam optimizer (Diederik, 2014) for all methods. Given that edits in our setup are single and sequential, the batch size is consistently 1. We trained all methods using a variety of GPUs, including 24GB NVIDIA RTX A5000s, 40GB NVIDIA A100s, and 80GB NVIDIA A100s. Timing experiments are reported from experiments performed on an NVIDIA RTX A100 GPU. The scale of BalancEdit is not dependent on the model's scale, but the model's scale is dependent on the available computational resources. To avoid sharding, we utilize models that can be accommodated on a single GPU, although the principles of BalancEdit are applicable beyond this setup. For Adaptor-based editors, such as GRACE, we employ 100 iterations of gradient descent per input.

## G. Key Distribution

To verify the key distribution in sequential editing, we present the distributions of keys in the codebook. From the Figure 7, we observe that the keys are scattered, indicating that the codebook is capable of handling multiple edits well.

