# OpenReview forum: "BalancEdit: Dynamically Balancing the Generality-Locality Trade-off in Multi-modal Model Editing"
_ICML.cc/2025/Conference — ICML 2025 poster_

### Official Review · Reviewer_D7mY · 2025-02-28

**Overall Recommendation:** 4

**Summary:**

The paper introduces BalancEdit, a method designed to address the challenge of balancing generality and locality in multi-modal model editing. Existing methods often fail to dynamically adjust the influence scope of edits, leading to over-correction or under-correction. The authors introduce the generality-locality trade-off in multi-modal editing and create OKEDIT, a dataset to evaluate this balance. Then, the proposed solution, BalancEdit, integrates an adapter into a vision-language model layer without altering the original weights. This adapter functions as a codebook that stores edits by caching input-error embeddings and updated transformation layers. Experiments show it outperforms baselines (e.g., IKE, MEND, GRACE) across metrics, achieving state-of-the-art results in targeted updates.

**Claims And Evidence:**

Strengths:
* BalancEdit balances generality and locality better than baselines.
* Efficiency of BalancEdit: Comparison of computational costs and data requirements strengthens this claim.
* Support for sequential edits.

Weaknesses:
* **Superiority of OKEDIT dataset**: While OKEDIT is described as more comprehensive than MMEDIT (table 2), details about its construction (e.g., GPT-4-generated rephrasings, diffusion-model-generated images) raise questions about potential biases. No qualitative examples of "harder" locality samples are provided compared with MMEDIT. In addition, the lack of dataset analysis for human-annotated benchmarks weakens its validity as a gold label.
* **Dynamic influence radius mechanism**: The radius calculation (Eq. 3) depends on hyperparameter $\alpha$ and distances between positive/negative samples. While ablation studies show α’s impact, the paper does not explain how α is chosen or whether it generalizes across tasks.
* **Negative sample selection**: Although the author compares the Negative Anchor of white and black, I don't understand why a pure colour is used as a negative sample, rather than a counterfactual example that is then labeled very differently.

**Essential References Not Discussed:**

No.

**Experimental Designs Or Analyses:**

* Why doesn't time spent fine-tuning count as time spent doing training in tabel 5?
* I am more interested in knowing the extremes of hyperparameter $\alpha$ in ablation experiments, e.g. 0 or 1.

**Methods And Evaluation Criteria:**

* **Layer Selection**: The choice of layer (e.g., specific transformer blocks) is critical but not rigorously justified. Performance might degrade if suboptimal layers are selected.

* **Simplistic Positive Samples**: In addition to the negative samples mentioned above, the positive samples are simply perturbed by simple textual. Is it possible to retrieve associated images and use them as positive samples?

* The loss function in Eq. 2 is not clear.

**Other Comments Or Suggestions:**

Please see above.

**Other Strengths And Weaknesses:**

Please see the methods and evaluation criteria.

**Questions For Authors:**

Please see above.

**Relation To Broader Scientific Literature:**

The paper’s key contributions are closely tied to and advance several strands of prior work in model editing, multi-modal learning, and parameter-efficient adaptation.

**Theoretical Claims:**

There seems to be no issue with the theoretical part.

---

> ### Author Rebuttal · Authors · 2025-04-01
>
> Thanks a lot for the valuable feedback! We would like to clarify some points below.
> >Why our locality sample is harder?
>
> A: Thanks for the question. Our locality sample is harder due to the **sematical similarity** to the editing knowledge. For example:
>
> |Aspect|MMEDIT|Our (OKEDIT)|
> |-|-|-|
> |**Edit Input**|Q: How many tennis balls are in the picture? A: 0 → 2| Q: What brand is this computer? A: HP → Lenovo|
> |**Locality Sample**|Q: What sport can you use this for? Image: bicycle   A: riding|Q: What brand is this computer? Image: Dell laptop   A: Dell|
> |**Relation to Edit**|Semantically **unrelated**|Semantically **similar**|
> |**Difficulty**|Low：easy to distinguish|High: requires finegrained reasoning|
>
> We will add more qualitative comparisons in the paper.
> >How is the validity of the dataset?
>
> A: Thanks for raising this important point. We provide the **Category-level Statistics** of OKEDIT in our response to Reviewer 55Ec, showing its generality. We also performed **human verification** on a subset of samples to ensure the correctness and the alignment with the intended edit scope. This helps establish the reliability of OKEDIT.
> >How is α selected?
>
> A: Thanks for the valuable concern. We select α using a small **held-out set** of only 5 unrelated samples. Since different models may have different latent feature distributions, α is treated as **model-dependent**. Once chosen, the same α is fixed per model (e.g., α=0.2 for MiniGPT-4) for all evaluations.
> >Why is our negative sample better than generating counterfactual negative sample.
>
> A: Thanks for the thoughtful question. We want to clarify that using counterfactual samples presents several limitations:
> - **No Extra Knowledge Assumption:**
> We aim to **avoid requiring additional knowledge** beyond the editing input for our goal of being **realistic and efficient**, but counterfactual sample require external knowledge.
> - **Intractability of Counterfactuals:**
> Generating high-quality counterfactuals is costly, while ours use an **efficient, universal, and reusable** negative anchor.
>
> To further justify the effectiveness of our negative anchors, we conducted an empirical comparison with a **random negative sample baseline** (in the second last response to reviewer ZhWy). We still achieve better results.
> >How is the editing layer chosen?
>
> A: Thanks for raising this concern. In our method, we select the editing layer based on prior work(MEND, MMEDIT).
>
> To validate the robustness of our method, we conducte an ablation study on another editing layer below:
>
> |Method|Acc|T-Gen|I-Gen|Loc|HM|Model|
> |-|-|-|-|-|-|-|
> |BalancEdit|100|99.87|76.46|53.14|71.58|MiniGPT4|
> |diff layer|100|100|64.03|66.39|73.75||
> |BalancEdit|100|98.89|65.38|61.18|71.85|BLIP2|
> |diff layer|100|100|78.43|44.99|66.70||
>
> Even if a different layer is chosen, **we can still achieve good performance**, showing the robustness of our method.
> > Why is your positive sample better than retrieving associated positive images?
>
> A: Thanks for the suggestion. We would like to clarify that retrieval would introduce **additional overhead**, including retrieval pipelines and sample bank. Retrieval also assumes that **visually similar positive samples exist**, which may not hold for long-tail or rare edits.
>
> In contrast, our method uses **textually rephrased questions**, enabling **efficient and consistent construction** of positive samples, keeping our framework lightweight and generalizable across edits.
> >The loss function in Eq. 2 is not clear.
>
> A: Thanks for pointing this out. $L$ is the **language loss**, specifically the **next-token prediction loss** used by the base LLM. This loss ensures that the updated model $f_{\text{new}}$, when processing input$(i, t)$, generates the desired new answer $y_n$.
> >What does the time in table 5 mean?
>
> A: Thanks for the question. In Table 5, **“training” refers to the offline pretraining phase**, such as training a metanet (e.g., MEND). However, We don't need that.
>
> The **time spent fine-tuning in our method is counted under "editing time"**, as it is performed **per edit**. The fast edit time shows that our method is **efficient**.
> >What is the extreme situation for α?
>
> A: Thanks for the suggestion. We conduct ablation experiments using extreme values of α on Minigpt4 with OKEDIT dataset:
>
> |Method|Acc|T-Gen|I-Gen|Loc|HM|
> |-|-|-|-|-|-|
> |BalancEdit|100|99.87|76.46|53.14|71.58|
> |α=0|100|100|95.19|16.30|36.65|
> |α=1|100|45.94|24.22|100|41.06|
>
> - **When α = 0**, the influence radius is entirely determined by the **negative sample**, leading to **over-generalization and lower locality** (e.g., 16.30 Loc), as the edit is applied too broadly.
> - **When α = 1**, the radius is determined by the **positive sample** only, leading to **over-localization and poor generalization** (e.g., 24.22 I-Gen), since the edit applies to narrow a region.
>
> These results confirm that α controls the trade-off between generality and locality as intended, which validates our assumption.

---

### Official Review · Reviewer_TTyf · 2025-03-13

**Overall Recommendation:** 4

**Summary:**

Large-scale multimodal models suffer from knowledge decay as facts change, and traditional fine-tuning is often impractical due to their size. Instead, direct knowledge editing is preferred, but current methods neglect differences in fact influence, creating a trade-off between generality and locality. To address this, the authors introduce the generality-locality trade-off, propose the OKEDIT dataset for evaluation, and present BalancEdit—a method that dynamically balances these aspects by generating positive and negative samples and editing the model’s latent space via a discrete, localized codebook without altering the underlying weights.

**Claims And Evidence:**

The claims are supported by the experiments.

**Essential References Not Discussed:**

No.

**Experimental Designs Or Analyses:**

The experimental design is valid.

**Methods And Evaluation Criteria:**

The proposed method make sense for the problem.

**Other Comments Or Suggestions:**

See weaknesses.

**Other Strengths And Weaknesses:**

Strengths:

Introduces the generality-locality trade-off, a unique perspective in multimodal model editing that explicitly addresses the balance between global consistency and localized accuracy.

The creation of the OKEDIT dataset provides a targeted benchmark for evaluating this trade-off, which can facilitate further research in the area.

BalancEdit’s mechanism of generating both positive and negative samples to determine the influence scope is a sophisticated approach that enhances the precision of edits.

Weaknesses:

The success of the method hinges on the quality and comprehensiveness of the OKEDIT dataset; any limitations or biases in the dataset could  affect the evaluation and generalization of the results.

**Questions For Authors:**

The multimodal large models used in the experiments are somewhat outdated. It is recommended to conduct experiments on the latest multimodal large models, such as llava-onevision, Qwen2-VL, and others.

**Relation To Broader Scientific Literature:**

There is a lot of related work; it is recommended that this paper clearly highlights the differences and contributions compared to existing multimodal knowledge editing works.

**Theoretical Claims:**

This paper do not contain theoretical claims.

---

> ### Author Rebuttal · Authors · 2025-04-01
>
> We thank the reviewer for providing valuable feedback to improve our paper. We have addressed your hesitations below.
> > Please highlight the contributions compared to existing works.
>
> A: Thank the reviewer for the valuable suggestions. We agree that clearer differentiation is necessary. Below, we restate our key contributions while explicitly contrasting them with existing multimodal knowledge editing works:
>
> 1. **We formulate the generality-locality trade-off in multi-modal model editing.**
>    In contrast, prior works such as MMEDIT [1] and GRACE [2] focus either on general-purpose editing or lifelong editing, but **do not explicitly define or evaluate** the trade-off between generality and locality in the multi-modal setting.
>
> 2. **We introduce OKEDIT, a new benchmark dataset designed to evaluate both generality and locality.**
>     In contrast, MMEDIT [1] uses random choose pairs as locality samples which may result in loose alignment and limited coverage.
>
> 3. **We propose BalancEdit, an efficient editing framework that dynamically adjusts the influence scope of edits.**
>    In contrast, IKE [3] assumes a wide influence range due to retrieval-based prompting, while GRACE [2] uses fixed-radius memory lookups. We introduce a radius-based mechanism using positive and negative samples to support **dynamic control over edit scoping**.
>
> 4. **We design a parameter and data efficient mechanism with little training overhead per edit.**
>    In contrast, MEND [4] and similar meta-learning approaches require pretraining on large edit datasets, which are hard to obtain in the multi-modal domain. BalancEdit **avoids pretraining**, supports multi-edit scenarios.
>
> We will revise the Introduction and Related Work sections to make these distinctions clearer in the final version.
>
> [1] Cheng et al., Can We Edit Multimodal Large Language Models?, arXiv 2023
> [2] Hartvigsen et al., Aging with GRACE, NeurIPS 2024
> [3] Zheng et al., Can We Edit Factual Knowledge by In-Context Learning?, arXiv 2023
> [4] Mitchell et al., MEND: Model Editing with Noisy Demonstrations, NeurIPS 2021
>
> > How is the quality and comprehensiveness of the OKEDIT dataset?
>
> A: Thank you for the important observation. While OKEDIT is our main evaluation benchmark, we have taken several steps to ensure the **robustness, diversity, and generalizability** of both the dataset and our method:
>
> 1. **Cross-dataset Validation:**
>    We evaluated BalancEdit not only on OKEDIT but also on the **MMEDIT dataset**, and observed consistently strong performance (see Table 3), which supports the **generalizability** of our method across datasets with different construction paradigms.
>
> 2. **Grounding in a Verified Source Dataset:**
>    OKEDIT is built upon the **OKVQA dataset**, which has been widely used and shown to exhibit **limited bias**. This provides a reliable and diverse base of real-world image-text pairs for constructing edits.
>
> 3. **Diverse Category Coverage:**
>    Our dataset spans **11 broad knowledge categories** (e.g., animals, transportation, brands, people, holidays, etc.), ensuring that our benchmark reflects both **common and long-tail factual knowledge**. This promotes **balanced evaluation** across a wide range of input types.
>
> 4. **Harder and More Realistic Locality Samples:**
>    Compared to MMEDIT, OKEDIT constructs **semantically similar locality distractors**, making the evaluation more challenging and reflective of **real-world edit interference scenarios**.
>
> 5. **Human Verification and Quality Control:**
>    We conducted **human validation** on a subset of the dataset to ensure correctness of labels and alignment with the intended edit scopes.
>
> Together, these points demonstrate that OKEDIT is a **carefully designed and reliable benchmark**, and that BalancEdit is not overly dependent on a specific dataset structure. We will revise the manuscript to better highlight these points.
>
> > How is the generalization of the method on newer model?
>
> Thank the reviewer for the constructive suggestion. To evaluate the generalization of our method on more recent vision-language backbones, we conducted additional experiments using **Qwen-VL** on the MMEDIT dataset. The results are shown below:
> |Method|Acc|T-Gen|I-Gen|Loc|HM|
> |-|-|-|-|-|-|
> |Base|18.18|17.13|14.14|NA|NA|
> |GRACE|99.88|30.36|32.18|87.78|39.78|
> |BalancEdit|100.00|70.96|71.51|41.96|**57.79**|
>
> Our method achieves a **higher harmonic mean (HM)** compared to GRACE, indicating a **more balanced trade-off between generality and locality**. These results also validate the **generalizability** of BalancEdit to newer backbone models beyond those originally reported in the paper. We will add more backbones in the future work.

---

### Official Review · Reviewer_ZhWy · 2025-03-13

**Overall Recommendation:** 2

**Summary:**

Existing multi-modal model editing methods struggle to dynamically adjust the influence scope of an edit, balancing generality and locality.
To address the issue, this paper proposes a novel model editing method, i.e., BalancEdit, with process as follows

* For each image-text pair to be edited, the averaged embedding and the associated transformation and radius are cached.
The transformation is learnt by standard fine-tuning and the radius is an interpolation of distances with positive (rephrased text, same image) and negative (same text, black image) samples.
* For new image-text pair, an edited transformation is activated if it falls within the scope of an edited sample; otherwise, the unedited transformation is applied.

The proposed method is empirically compared with FT, IKE, MEND, and GRACE on MMEDIT and OKEDIT datasets in both single and sequential editing settings, where BalancEdit enjoys superior accuracy, text generality, harmonic mean, and efficiency.
Ablation study is conducted to illustrate the effects of hyper-parameters, distance function, and anchors.

**Claims And Evidence:**

The author claims that BalancEdit is parameter-efficient on line 108, while I understand that it requires to cache a transformation for each edit, making it suffers linear memory consumption with respect to the number of edits.

**Essential References Not Discussed:**

NA

**Experimental Designs Or Analyses:**

See Methods and Evaluation Criteria

**Methods And Evaluation Criteria:**

I understand the intuition behind the algorithm while I am curious about whether the design is sufficiently justified.

* The averaged embedding is chosen as the cached key.
I wonder whether there is any reference or experiment to support the choice.
Will it be better than the embedding of certain influential tokens discovered by prior work [1]?
* The same text with black image is selected as the negative sample.
Is there any empirical study to demonstrate its superiority to a straight-forward baseline, i.e., randomly choosing irrelevant text-pair samples?

[1] Locating and Editing Factual Associations in GPT, NeurIPS 2022.

**Other Comments Or Suggestions:**

* No period on line 87.
* The Metric paragraph in Sec. 4.1 seems should be separated.

**Other Strengths And Weaknesses:**

* The idea of the work makes sense and is interesting to me.

**Questions For Authors:**

NA

**Relation To Broader Scientific Literature:**

NA

**Theoretical Claims:**

This work is mostly empirical.

---

> ### Author Rebuttal · Authors · 2025-04-01
>
> We thank the reviewer for the insightful feedback, we are glad for the opportunity to clarify some points.
> >Why is our method parameter-efficient?
>
> A: Thanks for the valuable comment! Knowledge editing is a challenging task, and we acknowledge that memory grows linearly with the number of edits due to cached transformations. However, compared to baselines like full fine-tuning or meta-learning (e.g., MEND), BalancEdit is still parameter-efficient per edit, as it **only** modifies a single layer and avoids retraining the full model.
> Moreover, our transformation can be replaced by PEFT method (e.g., LoRA), which would further reduce memory usage. We chose full fine-tuning for **universality** and for **clearer and fair comparisons**.
>
> >Why use averaged embedding as the key?
>
> A: Thank the reviewer for the thoughtful question. We would like to clarify that using the **averaged embedding as a sentence-level representation** is a common and effective strategy(e.g., SimCSE [1], SBERT [2]), and we adopt a similar approach to cache edited knowledge in BalancEdit.
>
> We chose this method over influential-token-based strategies (e.g., ROME [3]) for two main reasons:
>
> 1. **Modality Gap:** Prior work like ROME is designed for *text-only* models (e.g., GPT), while we focus on **multi-modal** models, where token-level attributions are harder to isolate due to vision-language fusion.
>
> 2. **Simplicity and Generalization:** Unlike token-attribution methods, our approach does not require edit-specific or model-specific token selection, making it easier to generalize across diverse edits and architectures.
>
> We find that average embedding is robust, simple, and effective for our editing framework.
>
> [1] Gao et al., SimCSE, EMNLP 2021
> [2] Reimers & Gurevych, Sentence-BERT, EMNLP 2019
> [3] Meng et al., Locating and Editing Factual Associations in GPT, NeurIPS 2022
>
> >Why does your Negative sample selection better than randomly chosen negative sample?
>
> A: Thank the reviewer for the insightful question. We conducted an empirical comparison with a **random negative sampling baseline**, where a randomly chosen text-image pair (assumed irrelevant to the edit) is used as the negative anchor.
>
> | Method                | Edit Acc | T-Generality | I-Generality | Locality | HM    | Model    |
> |-----------------------|----------|--------------|--------------|----------|--------|-----------|
> | **BalancEdit**        | 100.00   | 98.89        | 65.38        | 61.18    | **71.85** | BLIP-2    |
> | Random Negative Sample| 100.00   | 100.00       | 49.12        | 65.08    | 65.61  | BLIP-2    |
> | **BalancEdit**        | 100.00   | 99.87        | 76.46        | 53.14    | **71.58** | MiniGPT-4 |
> | Random Negative Sample| 100.00   | 99.00        | 66.93        | 45.92    | 64.08  | MiniGPT-4 |
>
> **BalancEdit consistently outperforms the random baseline** in harmonic mean (e.g. 71.85 vs. 65.61), showing a better balance between generality and locality. This supports the effectiveness of our **black image-based negative anchor**, which provides a **fact-agnostic, consistent, and efficient way to define a lower bound** in representation space.
>
> In contrast, random negative samples:
>
> - Depend on external unrelated examples,
> - Are unstable in quality and relevance,
> - And require assumptions that may not hold across diverse domains.
>
> Our method avoids these issues, making it more **robust and scalable** for real-world editing scenarios. We will include this analysis and clarify the design rationale in the final version.
>
> >Typos:
>
> A: Thanks for the detailed review. We will revise them accordingly.

---

> > ### Comment · Reviewer_ZhWy · 2025-04-06
> >
> > > Parameter-efficient
> >
> > The author points out that the proposed method is more parameter-efficient than fine-tuning and meta-learning.
> > I deem that the tuned parameters in fine-tuning and meta-learning are hyper-parameters (that can be set to a single linear layer), thus they are not inherently more expensive than the proposed method.
> > Perhaps a more suitable claim is that the proposed method achieves a better performance under limited budgets (where only a single linear layer is tuned).
> > I am still skeptical of the claim regarding parameter efficiency. When editing thousands of knowledge, the additional parameters may (significantly) exceed the original parameters of the model.
> >
> > > Average embedding
> >
> > I appreciate the explanation.
> > I understand that the average embedding of SimCSE and SBERT are tuned as representation of the sentence in contrast to the zero shot setting in the proposed method, so they may not well support the choice.
> >
> > > Random negative
> >
> > I appreciate the author for the additional experiment as I deem it is necessary to show the proposed method at least outperform a straight-forward baseline.

---

> > > ### Author Response · Authors · 2025-04-07
> > >
> > > Thank you for your reply! We would like to clarify it below.
> > >
> > > >Parameter-efficient
> > >
> > > A: Thank you for the thoughtful feedback. We agree a more accurate framing is that BalancEdit **achieves strong performance under constrained tuning budgets**, particularly when compared to full fine-tuning or meta-learning-based methods. We appreciate the opportunity to clarify our claim on efficiency as follows.
> > >
> > > 1. **Scalability via edit merging:**
> > >    To address cumulative cost when editing **many knowledge points**, we design a **merging mechanism for sequential edits** (Section 3.2 line 200). When edits occur in a coherent region of the latent space, they can be **consolidated into a single transformation**, reducing storage and memory overhead. We will further highlight this mechanism and its benefits in the final version.
> > > 2. **No pretraining or shared meta-parameters:**
> > >    Unlike existing methods such as MEND, BalancEdit avoids **offline meta-learning**, large shared modules, or retraining on auxiliary datasets. Edits are fully modular and isolated, and thus is efficient per-edit.
> > >
> > > We will revise the efficiency claims as suggested and add more discussions on cumulative cost and merging in multi-edit scenarios.
> > >
> > > >Average embedding
> > >
> > > A: Thank you for the thoughtful follow-up. We agree that SimCSE and SBERT are tuned for sentence representation. In fact, we were trying to clarify that **mean pooling can be effectively employed to aggregate contextual information across all tokens**, as demonstrated in related work and experiments [1–5], even from decoder-only models such as GPT or LLaMA, without any additional training.
> > >
> > > Our intention was not to claim that average pooling is the optimal strategy, but rather that it is a **commonly used and practical approximation** of sentence-level semantics, similar in spirit to approaches such as last-token pooling (e.g., GRACE) or influential neuron tracing (e.g., ROME). Compared to token attribution methods, our use of average embeddings is **model-agnostic, efficient, and requires no task-specific intervention**.
> > >
> > > Moreover, our experiments validate that this strategy is **effective in practice**, achieving strong performance across datasets, and achieving a balance between generality and locality.
> > >
> > > We will clarify this design choice and add the above references in the final version.
> > >
> > > [1] Tao et al., LLMs are Also Effective Embedding Models: An In-depth Overview, 2024
> > > [2] Su et al., One Embedder, Any Task: Instruction-finetuned Text Embeddings, ACL 2023
> > > [3] BehnamGhader et al., LLM2Vec: Large Language Models Are Secretly Powerful Text Encoders, 2024
> > > [4] Springer et al., Repetition Improves Language Model Embeddings, 2024
> > > [5] Lee et al., Gecko: Versatile Text Embeddings Distilled from Large Language Models, 2024
> > >
> > >
> > > > Random negative Experiment
> > >
> > > A: We truly appreciate your constructive feedback and will make sure to include this comparison in the final version.
> > >
> > > We sincerely appreciate the time you dedicated to the review. We hope that our response can address the concerns you have.

---

### Official Review · Reviewer_55Ec · 2025-03-13

**Overall Recommendation:** 2

**Summary:**

The paper introduces a method for efficiently updating multi-modal LLMs. Existing model editing struggles with balancing generality  and locality. BalancEdit addresses this by using a codebook mechanism that stores discrete edits, dynamically adjusting their influence scope with positive and negative samples. The authors introduce OKEDIT, a dataset designed to evaluate this trade-off. Experiments on MiniGPT-4 and BLIP-2 OPT show that BalancEdit surpasses baseline methods while maintaining efficiency and interpretability.

**Claims And Evidence:**

The claims in the submission are partially supported by empirical evidence, including quantitative evaluations, baseline comparisons, and ablation studies. The introduction of the OKEDIT dataset provides a structured way to assess the generality-locality trade-off, and experimental results on MiniGPT-4 and BLIP-2 OPT indicate that BalancEdit performs effectively in terms of accuracy, efficiency, and interpretability. Efficiency claims are supported by editing time comparisons.

**Essential References Not Discussed:**

NA

**Experimental Designs Or Analyses:**

The experimental design includes baseline comparisons, metrics (harmonic mean of generality and locality), and ablation studies to validate BalancEdit. The OKEDIT dataset serves as a relevant benchmark, while efficiency claims are supported by editing success and time analyses.

**Methods And Evaluation Criteria:**

The proposed methods and evaluation criteria are appropriate for multi-modal model editing, with BalancEdit addressing the generality-locality trade-off and OKEDIT providing a structured benchmark. The chosen metrics—editing success, generality, and locality—effectively assess model edits, and comparisons with baselines support the evaluation.

**Other Comments Or Suggestions:**

Please enlarge the font size in Figure 3 to improve readability.

**Other Strengths And Weaknesses:**

Strength
1. This paper proposed BalancEdit which dynamically optimizes the generality-locality trade-off without modifying model weights.
2.The proposed method outperforms baseline methods in accuracy, efficiency, and sequential editing while requiring minimal computational cost.
3.This paper discuss the radius of the influence scope, which is useful for editing-related research.

Weaknesses
1. The organization of the paper, especially the organization of the method section, is not clear enough, make it hard to follow. (Detailed examples are in Question section).
2.    It could be questionable how to ensure the quality of the generated images for generality and locality respectively, since sometimes there may not be a clear boundary between them. The test dataset is also built on generated data, which increases my concern.
3.    The reproducibility of the method is another concern.

**Questions For Authors:**

It is still difficult for me to understand what transformations (v) is.
It is hard to understand the eq. (2). And what is L in this equation?
What is a specific key k?
“By caching embeddings for input errors and the updated knowledge transformation layer that decodes into the desired model outputs”.. Could you explain it in detail?
Why hyperparameter /alpha can “adjust the distance” in eq. (3).  if I understand correctly, /alpha is used to adjust the weight of different distances instead of “adjusting the distance”.
How did you ensure the quality of the generated images for generality and locality respectively?
Could you provide the statistics of each categories (vehicles, people, etc.), the number of images under generality, locality?

**Relation To Broader Scientific Literature:**

LLM editing is a promising research direction for cost‑effective revision of large language models. It can further remove harmful or biased content that might persist despite standard safety alignment mechanisms.

**Theoretical Claims:**

I have checked the equation in the paper. There is no proof in the paper.

---

> ### Author Rebuttal · Authors · 2025-04-01
>
> We thank the reviewer for the valuable feedback. Here we're glad for the chance to clarify some points.
> >Quality of the generality and locality samples?
>
> A: We appreciate the reviewer’s thoughtful concern. To ensure the semantic quality of the OKEDIT dataset and the meaningful separation between generality and locality, we adopt a carefully designed generation pipeline. The detailed generation process has been provided in Appendix A.1. Here we provide a concrete example.
>
> The editing counterfactual knowledge is ***the "HP" looks computer is named as lenovo.*** Specifically, the original image is a HP brand computer, original question is ***'What is the brand of it'***, and the original result is ***'HP'***, and the new result is ***'lenovo'***.
>
> - **Image Generality sample generation:**
> 1. **Generality object confirmation:** We ask GPT4 about objects and scene that should be in the image. The prompt is shown in Appendix A.1. The GPT would answer ***"A HP laptop"***.
> 2. **Generality sample generation:** Based on the object, we will prompt the diffusion model to generate the images. Since we provide the exact object, the model can generate correct images.
> - **Image locality sample generation:**
>     1. **Locality answer generation**: We first prompt GPT4 to generate a distractor. Like *"Given (question: What is the brand of it, A: HP, B: Lenovo), what could be another option? Short answer. C: []"* The GPT would generate a distractor, such as ***"Dell"***
>     2. **Locality object confirmation:** Similarly, we ask GPT4 about objects and scenes that should be in the image. The GPT would answer ***"A Dell laptop"***.
>     3. **Locality sample generation:** Based on the locality object, we will ask diffusion model to generate the correct images.
>
> In this way, having clear instructions for creating an image helps ensure its quality.
> >What is transformation(v)? What is L and a specific key k.
>
> A: We apologize for the confusion about Eq. (2). In the following. We clarify each component of Eq. (2) and we will update the final version of our paper accordingly:
>
> **Transformation $v$**：The transformation refers to a **specific layer in the LLM**. During an edit, we **fine-tune this transformation layer** to encode the new knowledge. Thus, $v$ is a **updated version of an existing LLM layer**, reused only for inputs within the influence radius of a particular edit.
>
> **Loss $L$**: $L$ is the **language loss**, specifically the **next-token prediction loss** used by the base LLM. This loss ensures that the updated model $f_{\text{new}}$, when processing input$(i, t)$, generates the desired new answer $y_n$.
>
> **Key $k$**: As shown in Method 3.2, each **$k$** in the codebook is a **representative embedding** for a specific edit.
> For example, suppose we edit the fact: *"What brand is this computer?" from "HP" to "Lenovo".* Then the key k is the averaged embedding from layer $l-1$ when the input is the ***HP laptop image + the original question***.
> >“By caching embeddings for input errors and the updated knowledge transformation layer that decodes into the desired model outputs”.. Could you explain it in detail?
>
> A: Thank the reviewer for pointing this out. We agree that this sentence may have been too condensed, and we are happy to explain it in detail. Similar to the Section 3.2, We mean:
> 1. **Caching embeddings for input edits (keys $k$):**
>    For each edit (i.e., input $(i, t)$ and new answer $y_n$), we extract the **averaged embedding** from the layer $l-1$ of the model. This embedding — the **key $k$** — represents the semantic location of the edit in the model’s latent space. It serves as a **reference point** to decide whether future inputs are close enough to activate the edit.
> 2. **Caching the transformation $v$:**
>    We fine-tune and cache a **copy of the LLM’s layer $l$** so that, when applied to this type of input, it produces the new target output $y_n$.
>
> >How does α work?
>
> A: Thanks for mentioning the key point. α adjusts the **influence radius** by **weighting** the distances between the key and the positive/negative samples. It also controls the trade-off between generality and locality.
> >Statistics of categories
>
> A: Thank the reviewer for raising this point. Below, we provide detailed statistics of the knowledge categories in the OKEDIT dataset. As shown in the table, OKEDIT spans a **diverse and representative** set of domains, supporting its utility as a comprehensive and generalized benchmark for multi-modal model editing.
>
> |KnowledgeCategory|Percentage|
> |-|-|
> |Plants, Animals|17%|
> |Vehicles, Transportation|16%|
> |Cooking, Food|15%|
> |Sports, Recreation|12%|
> |People, EverydayLife|9%|
> |Objects,Material, Clothing|8%|
> |Brands,Companies, Products|3%|
> |GeographyHistory,Language, Culture|3%|
> |Weather, Climate|3%|
> |Science, Technology|2%|
> |Other|12%|
>
> >Reproduction and typos
>
> A: We thank the reviewer's attention. We will fix the typos based on the suggestions and provide the code in the final version.

---

### Decision · Program_Chairs · 2025-05-01

**Decision:**

Accept (poster)

**Comment:**

After the rebuttal and discussion phases, the paper received scores of 4, 4, 2, and 2, with an average rating of borderline accept. Most reviewers acknowledged the contributions of the paper, including its effective balance between global consistency and localized accuracy in model editing, the useful methodology and benchmark it provides for editing-related research, and its cost-effectiveness.

Two reviewers (Reviewer 55Ec and Reviewer ZhWy) did not provide sufficient responses during the rebuttal and discussion phases. After carefully reviewing the reviewers' comments, the authors' responses, and conducting my own brief examination of the paper, I believe that the major concerns raised by Reviewer 55Ec and Reviewer ZhWy have been adequately addressed in the rebuttal and discussion stages.

Since the paper introduces the generality-locality trade-off in multi-modal model editing for the first time, I believe it offers valuable insights to the model editing community. Therefore, I recommend acceptance of the paper.